# Low-FODMAP Diet for Irritable Bowel Syndrome: Insights from Microbiome

**DOI:** 10.3390/nu17030544

**Published:** 2025-01-31

**Authors:** Haoshuai Zhang, Qi Su

**Affiliations:** 1Microbiota I-Center (MagIC), Hong Kong SAR, China; 2Department of Medicine and Therapeutics, The Chinese University of Hong Kong, Hong Kong SAR, China

**Keywords:** irritable bowel syndrome, low-FODMAP diet, probiotics, gut microbiota, dietary personalization

## Abstract

Irritable bowel syndrome (IBS) is a prevalent gastrointestinal disorder characterized by chronic abdominal pain, bloating, and altered bowel habits. Low-FODMAP diets, which involve restricting fermentable oligosaccharides, disaccharides, monosaccharides, and polyols, have emerged as an effective dietary intervention for alleviating IBS symptoms. This review paper aims to synthesize current insights into the impact of a low-FODMAP diet on the gut microbiome and its mechanisms of action in managing IBS. We explore the alterations in microbial composition and function associated with a low-FODMAP diet and discuss the implications of these changes for gut health and symptom relief. Additionally, we examine the balance between symptom improvement and potential negative effects on microbial diversity and long-term gut health. Emerging evidence suggests that while a low-FODMAP diet can significantly reduce IBS symptoms, it may also lead to reductions in beneficial microbial populations. Strategies to mitigate these effects, such as the reintroduction phase and the use of probiotics, are evaluated. This review highlights the importance of a personalized approach to dietary management in IBS, considering individual variations in microbiome responses. Understanding the intricate relationship between diet, the gut microbiome, and IBS symptomatology will guide the development of more effective, sustainable dietary strategies for IBS patients.

## 1. Introduction

Irritable bowel syndrome (IBS) poses a significant challenge for individuals worldwide, manifesting as chronic abdominal discomfort, bloating, and irregular bowel patterns. In response to these distressing symptoms, low-FODMAP diets have emerged as a promising dietary intervention, focusing on limiting fermentable carbohydrates to alleviate IBS-related issues effectively. This review delves into the current understanding of how a low-FODMAP diet impacts the gut microbiome and its mechanisms in managing IBS. By exploring the changes in microbial composition and function associated with this dietary approach, we aim to shed light on its implications for gut health and symptom relief. However, while consuming a low-FODMAP diet shows promise in reducing IBS symptoms, concerns arise regarding its potential negative effects on microbial diversity and long-term gut well-being. We delve into strategies like the reintroduction phase and probiotic use to counterbalance these effects. Emphasizing the need for a personalized approach to dietary management in IBS, we underscore the significance of considering individual variations in microbiome responses. By unraveling the intricate interplay between diet, the gut microbiome, and IBS symptomatology, this review aims to pave the way for the development of more efficacious and sustainable dietary strategies tailored to IBS patients’ needs.

## 2. What Is IBS?

The term functional bowel disorders (FBDs) is used to describe a chronic disordered stomach and bowel function or work with an unclarified pathology of structure attributable to the middle or lower gastrointestinal tract. The clinical symptoms are primarily manifested as bloating, distention, bowel habit abnormalities, and unspecified functional bowel disorders [1]. FBDs can be categorized into five main types: irritable bowel syndrome (IBS), functional constipation (FC), functional diarrhea (FDr), functional abdominal bloating/distention, and unspecified FBDd. Functional gastrointestinal disorders encompass a spectrum of conditions, including functional constipation (FC), characterized by infrequent or difficult bowel movements and dry, hard stools often accompanied by a sense of incomplete evacuation and abdominal bloating. Functional diarrhea (FDr) presents with increased bowel movement frequency, watery or loose stools, urgency, and abdominal pain. Functional abdominal bloating/distention manifests as a sensation or appearance of abdominal bloating without necessary pain or bowel movement irregularities. These disorders, lacking discernible structural abnormalities, are diagnosed based on symptom profiles and after ruling out other organic pathologies. Among these, IBS is one of the most common types of functional bowel disorders. IBS is defined as a functional bowel disorder in which abdominal pain or discomfort is associated with defecation or a change in bowel habit and with features of disordered defecation [1]; it is also common after certain acute events, such as diverticulitis or bacteria gastroenteritis [2,3]. Postinfectious IBS (PI-IBS) is also a main type, which is mostly caused by water pollution and protozoan and helminth infection [4]. IBS clinical symptoms can usually be described as increases in bloating, flatulence, global symptoms, and relating to more changes in bowel habits, like constipation or diarrhoa [5]. IBS can be classified into four main subtypes by predominant stool pattern: IBS with constipation (IBS-C), IBS with diarrhea (IBS-D), mixed IBS (IBS-M), and unsubtyped IBS. Although IBS is not a life-threatening disease that does not cause permanent damage to the gastrointestinal (GI) tract, intestinal bleeding, or other serious complications like cancers, it is still a difficult curable and global common healthcare problem. Over the past few decades, the prevalence of IBS has increased; the global prevalence of IBS, according to the Rome III criteria [6], has increased by 9.2% from 2006 to 2019 [7]. Most studies show that the prevalence of IBS is approximately 10–15% within communities [8,9,10,11] in different areas of the world. As for the Asian region, the prevalence is different across Japan, China, and South Korea at 14.9%, 5.5%, and 15.6% [10], respectively. Furthermore, the IBS prevalence in Hong Kong is 6.6% based on the Rome II criteria [12], amounting to nearly 46,000 people. Studies also show that the prevalence is related to the factors of gender and age. IBS prevalence in women is significantly higher than it is in men around the world [13], and it is lower in those aged above 50 years compared with those aged less than 50 years [14]. IBS leads to severe impacts on government economic budgets and patients’ quality of life. In Western countries, patients with IBS account for 40–60% of referrals to gastroenterology outpatient clinics [15] in the UK, and the annual national healthcare costs associated with IBS amount to USD 1.66 billion in the USA [16]. Additionally, patients with IBS seem to have worse health-related quality of life (HRQoL) than patients with certain other conditions such as gastroesophageal reflux disease, diabetes, and end-stage renal disease [17]. For example, most IBS patients feel a lack of freedom and spontaneity, and they emphasize the unpredictability of their symptoms and express that they may feel stigmatized by and misunderstanding from their relatives or friends [18]. Former studies also show that patients with IBS find it difficult to work continuously because of the symptoms, although they have tried their best to perform better. Research from the USA suggests that over 5% of people with IBS require some time off work, which leads absenteeism, and as a result, more than 20% of patients felt an overall work productivity loss [19]. Consequently, IBS not only affects patients individually but also influences all of society; the rising prevalence of the disease is an urgent issue that needs to be addressed.

IBS is commonly contributed to the disorders of gut–brain interactions. Although the precious pathogenic mechanism of IBS is complex and incompletely clear, many possible risk factors have been identified, such as genetics, dietary factors, infection, gut microbiome, and psychological comorbidity (stress, anxiety, or depression). Current medical treatments of IBS are usually based on an individualized evaluation, explanation, and reassurance. Alterations in diet, drug treatment aimed at predominant symptoms, and psychotherapy may be beneficial. Traditionally, increasing intake of dietary fiber in the usual diet can effectively reduce IBS symptoms [20], mainly constipation. Psyllium is one representative fiber supplement; research has shown that it improved symptom severity in the third month and adequately relieved abdominal pain in the first and second months compared with a placebo [21] in a clinical trial. If psyllium treatment fails, drug treatment is used. Drugs that are primarily used to alter adverse bowel habits and to alleviate the symptoms of bloating and abdominal pain mainly included antispasmodics (dicyclomine), laxatives (magnesium salts), anti-laxatives (loperamide), antidepressants [22], and non-absorbed antibiotics (rifaximin) [23]. Despite these treatment options, eradication therapy is also commonly used, which involves using a combination of proton pump inhibitors (PPIs) and two antibiotics, such as amoxicillin and clarithromycin. Previous research data have supported the suggestion that of 8 out of 204 IBS patients treated with 200 mg of rifampicin four times a day for 1 month, symptom scores improved significantly in 7 of the 8 treated with antibiotics [24]. Prebiotics have also been used in recent decades [25], including the usage of single-or multi-strain probiotics [26] comprising *Lactobacillus* and *Bifidobacterium* [27]. The use of prebiotics is a potentially promising approach that involves altering the gut microbiota and alleviating the symptoms of FBDs [28,29]. As alluded to earlier, fiber supplementation and drug treatments can release IBS patients’ symptoms, and the efficacy of eradication therapy is also effective. However, the effectiveness of fiber supplements is limited and can only improve symptoms of diarrhea. Similarly, the overuse of antibiotics is unhealthy for the host, and it may lead to antibiotic resistance. The emergence and spread of drug resistance impact not only the host’s therapy but can also lead to other severe diseases and public health issues. Thus, it is urgent to develop a more effective and potentially beneficial method of treating IBS symptoms that can actually solve the problem.

## 3. Relationship Between Gut Microbiota and Low-FODMAP Diet

Most animals coexist with prokaryotic symbionts, which provide various physiological benefits. The human gastrointestinal (GI) microbiota has an important role in human health. Millions of microbial communities colonize the small intestine, including bacteria, fungi, viruses, protozoa, and archaea, offering essential metabolic and immunological feelings and protective functions that are vital for human health [30]. Both in terms of quantity and quality, the bacterial population in the human body is exceedingly vast. The bacteria, archaea, and eukaryotes that inhabit the gastrointestinal tract are defined as the “gut microbiota”. They have co-evolved with their host over thousands of years, establishing a complex and mutually beneficial relationship [31]. In terms of quantity, the number of bacteria in the human body is significantly vast and roughly equivalent to the number of human cells [32], and the colon bacterial community makes up a significant proportion of the total bacterial population, nearly 10–11 cells/g [33]. For multicellular organisms like humans, the primary function of the intestines is nutrient concentration and absorption, making it a key site and experimental ground for eukaryote–prokaryote interactions. The microbiota composition in the GI tract is reflective of the physiological properties in each region (Figure 1). The GI microbiota plays an important role in human health, and former studies have illustrated that the beneficial symbiotic bacteria that coexist long-term within the human body exhibit varying degrees of epithelial immune suppression properties [34] and diverse abilities to stimulate the development of immune-suppressive T cells [35]. This suggests a strong rationale for the presence of the gut microbiota, which plays a crucial role in maintaining the host’s health. Similarly, the gut microbiota provides many other benefits to the host, such as enhancing gut barrier function or building the intestinal lining [36], extracting energy [37], and defending against pathogens [38]. The composition of the gut microbiota is influenced by environmental and host selective pressure. Accordingly, the gut microbiota is flexible, and various kinds of factors, including, but not limited to, environmental conditions, dietary habits, and the host’s health status, can all affect the diversity of microbial species.

It is reported that the diet significantly influences the composition of the gut microbiota [39]. Hippocrates raised a concept called “let food be thy medicine and medicine be thy food”, which can explain the relationship between the diet and gut microbiota components clearly. The gut microbiota accompanies the host for their whole life, from the beginning to the end. As an infant, human milk oligosaccharides (HMOs) play a role in the development of the microbiota in early infancy [40], which is followed by an increase in bacterial diversity upon the introduction of solid foods [41]. This process culminates in a reduction in diversity seen in elderly populations residing in long-term care facilities, likely due to a decrease in dietary variety [42]. Nutrients ingested by the host can directly influence the types, abundance, and growth status of the intestinal microbiota, thereby affecting the relative and absolute abundance of different microbial communities in the gut. Therefore, investigating the effects of different dietary ingredients on the gut microbiota composition is crucial for evaluating the health benefits of specific habitual diets. Typically, small-intestinal bacterial overgrowth (SIBO) is common in IBS patients [32,43,44], where quality bacteria (aerobic and anaerobic) overcolonize in the gastrointestinal tract, leading to metabolic activities that produce numerous substances that the host fails to utilize, such as hydrogen, methane, carbon dioxide, and certain short-chain fatty acids. The accumulation of these substances in the small intestine result in symptoms such as excessive belching, flatulence, epigastric and abdominal pain, nausea, early satiety, and fatigue [45], thus leading to altered bowel habits. These manifestations can subsequently contribute to the development of IBS, causing significant distress for patients. Based on the above reasons, a new hypothesis called FODMAP [46] was raised, which represents fermentable oligosaccharides, disaccharides, monosaccharides, and polyols. Typical FODMAP foods include fructose, fructans, lactose, sugar alcohols, and galacto-oligosaccharides (GOSs). The effects of FODMAP foods in the gastrointestinal tract are exerted not only via fermentation but likely also through alterations in the microbiota, metabolome, permeability, and intestinal immunity (Figure 2). All these substances contain a significant quantity of sugar. Studies have revealed that a high-sugar diet can reduce the diversity of gut microorganisms [47]. Furthermore, FODMAP foods induce a variety of short-chain fatty acids and organic acids [48], which are rapidly fermented, leading the expansion of small-intestinal bacterial populations, which has been clearly demonstrated in both ileostomy recovery and magnetic resonance imaging (MRI) studies. There has been increasing interest in low-FODMAP diets in terms of treating IBS, and it is time to understand how the diet plays a significantly important role in shaping the intestinal microbiome.

## 4. Low-FODMAP Dietary Efficacy and Clinical Evidence

Consumption of a low-FODMAP diet leads to a considerable reduction in the intake of fiber, fructans, micronutrients, and GOS [49]. Following a low-FODMAP diet means a decrease in the intake of highly fermentable but poorly absorbed short-chain carbohydrates (SCCs) and polyols in daily life in order to relieve painful symptoms. To date, studies have mainly focused on how a low-FODMAP diet improves disease symptoms by altering the permeability of the intestinal epithelial barrier [50,51,52,53], the mechanism by which a low-FODMAP diet influences GI microbiome components, and how the effects on the disorder and patient health conditions might also be important. Diet therapy has already been used a basic method for improving IBS patient symptoms. “The traditional IBS diet” suggested by the National Institute for Health and Care Excellence (NICE) guidelines advises the importance of a healthy diet, nutrition, and lifestyle [54] with a proper eating gap time, which involves spacing out meals appropriately, drinking water instead of soft drinks, alcohol, and caffeinated beverages, reducing the intake of fats and reheated foods, and avoiding the excessive consumption of high-fiber foods such as whole-meal breads, bran-rich foods, and brown rice, through all of which individuals can significantly improve their digestive health and overall well-being.

Notably, a low-FODMAP diet represents the dietary restriction of FODMAP foods, which is recommended in treating patients with IBS. A significant number of randomized and controlled trials (RCTs) have been carried out to show the priority of the died in IBS patient symptoms relief. In 2008, the first study demonstrating a link between dietary FODMAPs and symptoms found that IBS patients were more likely to experience GI symptoms after the blinded consumption of escalating doses of fructose or fructans than after consuming glucose [55]. After that, an increasing amount of evidence that supports the effectiveness of a low-FODMAP diet in patients with IBS symptoms has been obtained. US research groups recruited clinical patients and designed comparative effectiveness trials comparing a low-FODMAP diet versus usual dietary recommendations in IBS patients with diarrhea (IBS-D). The results [56,57] showed that a significantly higher percentage of people in the low-FODMAP diet group compared to the standard dietary recommendations group experienced relief from abdominal pain and bloating, two of the most troublesome symptoms associated with IBS. Furthermore, notable enhancements were observed in terms of stool consistency, frequency, and urgency when compared to standard dietary recommendations for IBS. Improvements in quality-of-life (QoL) indicators and reductions in anxiety were also notable in the low-FODMAP diet group compared to the standard dietary recommendations group for IBS. A research group in Canada [58] also found that IBS symptoms were linked to dietary FODMAP content and associated with alterations in the metabolome among 37 patients with IBS; over a 3-week duration, 19 patients were treated with a low-FODMAP diet, while the other 18 patients were treated with a high-FODMAP diet. Moreover, despite trails which have identified the role of FODMAP foods in alleviating IBS symptoms, the impact of the presence or absence of gluten in a low-FODMAP diet is also vital. Therefore, a double-blind, placebo-controlled, randomized trial with 49 IBS patients was carried out in 2021 [59]. Patients were divided into two groups, where the intervention group received 5 gr/day of gluten powder with a low-FODMAP diet, while the placebo group received 5 gr of rice flour as a placebo with a low-FODMAP diet. The results indicated that in individuals with IBS, symptom exacerbation following the consumption of gluten-containing foods is solely attributed to the presence of fructans in the food, while gluten is accountable for symptoms in only a small percentage of patients. Different regions might have various dietary habits, and a low-FODMAP diet continues to demonstrate significant advantages in improving IBS symptoms compared to traditional diets in many regions [46,60,61,62,63,64]. Among several of studies, the comparative study on Australian traditional diets versus low-FODMAP diets stands out as the most representative [65]. The results demonstrated that IBS patients treated with a low-FODMAP diet had significantly reduced GI symptoms compared with patients consuming a typical Australian diet or subjects consuming their own habitual diet, such as abdominal pain and bloating. In recent years, many research organizations [66] have suggested that a low-FODMAP diet can be considered a first-line therapy for treating IBS based on six different random clinical trials’ [58,65,67,68,69] analyses. Although all six trials have faced varying degrees of criticism regarding placebo selection, the number of study participants, the success rate of blinding, and the duration of the intervention, the consistent results indicate the positive nature of this diet. Empirical evidence for following a low-FODMAP diet has confirmed the results of randomized studies, with approximately 70% of patients showing a response. Compared with other dietary strategies, a low-FODMAP diet still holds significant advantages. In a randomized controlled trial involving 84 diarrhea-predominant IBS patients in the USA [64], a comparison was made between a NICE dietary strategy [54] and a low-FODMAP diet. Although the primary endpoint of overall symptom relief was not achieved, individual symptoms showed both clinical and statistical improvements on the low-FODMAP diet. Results have also shown that a low-FODMAP diet had increasing advantages compared to a NICE diet in a study conducted in the UK. Overall, while there is currently no definitive evidence proving that a low-FODMAP diet can be used as a first-line treatment for IBS, the majority of patients experience some degree of improvement in GI symptoms after following a low-FODMAP diet in the short term. This approach has gained support and recommendations from many nutritionists. Low-FODMAP dietary therapy shows promising potential for further development and growth.

## 5. Mechanisms by Which a Low-FODMAP Diet Alleviates Symptoms and Restrictions

FODMAPs are a group of carbohydrates that are poorly absorbed in the small intestine and that are subsequently fermented in the small or large intestine. However, not all FODMAPs exacerbate symptoms in IBS patients. There are two main mechanisms through which FODMAPs trigger symptoms in IBS patients. Firstly, FODMAPs are poorly absorbed in the small intestine and have osmotic properties [48,70], leading to a net fluid secretion into the small intestine. This can cause distention or significant fluid accumulation in the small intestine, resulting in abdominal symptoms, and it can also increase the amount of fluid transported to the colon. Additionally, FODMAPs are rapidly fermented by colonic microbiota [67], leading to colonic distension due to gas production, accompanied by pain and bloating. A recent finding has revealed that a low-FODMAP diet can modify visceral hypersensitivity by enhancing colon microcirculation perfusion and reducing the expression of vascular endothelial growth factor [71]. Therefore, a low-FODMAP diet can prevent the scenarios mentioned above, leading to the suffering of IBS patients being alleviated and leading to an improvement in recovery rates and quality of life. Studies have illustrated that diet compounds can shape the gut microbiota in direct or indirect ways by affecting the host’s metabolism (Figure 3). For example, one previous study found that acute vitamin A deficiency in mice leads to an increase in *Bacteroides vulgatus*. This may occur because retinol inhibits the bacterium’s growth, possibly by reducing bile acids like deoxycholic acid that normally suppress it. Owing to the role of shaping the gut microbiota, many randomized, placebo-controlled RCTs have been designed to explore the different construction of the GI microbiota between a low-FODMAP dietary plan and traditional or sham dietary plans. In recent decades, numerous studies have continuously demonstrated the clinical effectiveness of a low-FODMAP diet in patients with IBS. All of these studies have illustrated that a low-FODMAP diet greatly improved discomfort in the GI tract and helped patients enhance their quality of life to a large extent [72]. One parallel, single-blinded, placebo-controlled trail recruited 52 patients from gastroenterology clinics in the UK, who were divided into two groups, namely, a control group treated with a normal diet and another group provided with a low-FODMAP diet designed by the same research dietician; the experiment lasted for 4 weeks. The results showed that over half of the patients who followed the low-FODMAP diet gained adequate relief of gut symptoms, together with a greater reduction in IBS severity scores. Studies [73,74,75,76] have shown that IBS patients treated with a low-FODMAP diet have a different gut microbiota composition compared with IBS subjects. A research group from the UK utilized metagenomics to determine high-resolution taxonomic and functional profiles of stool samples from IBS subjects and healthy people following their own diet. After 4 weeks of low-FODMAP diet treatment, the clinical response and microbiota species all changed [73], with an increase in beneficial gut bacteria (*Bacteroidetes*) and a decrease in harmful bacteria (*Firmicutes*). In addition, a low-FODMAP diet improves symptoms in IBS patients and may also operate through an alternative mechanism by altering the composition and relative abundance of intestinal microbiota in patients, leading to changes in metabolites and ultimately improving gut conditions. Given this, the intestinal microbiota also holds significant potential in the future for becoming biomarkers for IBS detection [77] and the effects of a low-FODMAP diet, and one England-based research team [78] has evidenced that metabolites can be seen as biomarkers to identify IBS subtypes.

Recently, more and more published well-designed clinical trials have shown that after 4-week low-FODMAP diet interventions, although intestinal symptoms improved to some extent, there was a significant alteration in the composition of the gut microbiota, particularly characterized by a decrease in the diversity and abundance of beneficial microbial populations, which is an unexpected but undeniable consequence. Many studies [78,79,80,81,82] involving diets for treating IBS suffer from the placebo effect, a limited duration, a lack of rigorous endpoints, a lack of randomization/blinding, and limited dietary assessments to confirm adherence. Moreover, different randomized, single-blind trails results have also revealed that a low-FODMAP diet resulted in a lower total bacterial load compared with traditional dietary habits or the dietary habits typical of some areas [80,83]. Multivariate modelling of fecal bacterial profiles of patients with IBS also shows that a low-FODMAP diet may have a significant impact on fecal bacteria [82], notably resulting in a reduction in fecal bacterial species [84,85], *Actinobacteria*, especially *Bifidobacterium* [84,86,87]. Studies show a clear decrease in the proportion and concentration of *Bifidobacteria*, with a lower absolute abundance also demonstrated. *Bifidobacteria* is a kind of Gram-positive bacterial that is strictly anaerobic, has fermentative rods, is often Y-shaped or clubbed as a final type, is one of the predominant bacterial communities in the human GI tract, and it plays a vital role in the GI tract health. Accordingly, among the proposed benefits of *Bifidobacteria*, the inhibition of enteropathogenesis and a reduction in rotavirus infection [88] are some of their best-established outcomes (Figure 4). Moreover, *Bifidobacteria* plays a crucial role in regulating the pH of the large intestine by releasing lactic and acetic acid [89], thereby inhibiting the proliferation of various harmful pathogens and putrefactive bacteria. Numerous studies have also demonstrated that *Bifidobacteria* can inhibit pathogens through the production of organic acids [90], antibacterial peptides [91], quorum-sensing inhibitors [92], and immune stimulation, among other mechanisms, providing molecular clues as to their capacity to prevent certain infections. In other words, the reduction in or disappearance of *Bifidobacteria* in the human intestine would indicate a harmful state. While a low-FODMAP diet has shown significant effects in most IBS patients after a period of treatment, the restrictive diet still induces alterations in the intestinal microbiota [66,93] to some extent, posing a potential threat to the host’s health. Therefore, there is still substantial potential for improvement in low-FODMAP diets to enhance their clinical applicability.

## 6. The Importance of FODMAP Personalization in Clinical Practice

Despite an increasing number of studies indicating that a low-FODMAP diet might be the most effective dietary intervention for IBS and the fact that it has been recommended as a second-line treatment option [94], its negative effects—specifically, the significant reduction in the abundance of *Bifidobacterium* [95,96] in the gut—remain a concern. Furthermore, most clinical trials [97] of low-FODMAP diets have evaluated short-term clinical endpoints (≤12 weeks), and the durability of the diet over the longer term is arguably more meaningful given the chronicity of symptoms of IBS. Existing research also indicates that this method might also reduce the host’s intake of calcium [86]; consequently, this could increase the risk of fractures and osteoporosis in affected individuals. Additionally, numerous studies show that the majority of patients with IBS (70–89%) report that specific foods exacerbate symptoms, and consequently, many patients limit or exclude some food items [60]; therefore, standard low-FODMAP diets may not be suitable for every patient. More recently, Staudacher HM and his team [98] enrolled 18 IBS patients who participated in a short-term clinical trial and went through structured FODMAP restriction, reintroduction, and personalization to join a long-term study over 12 months. The results indicate that nearly 70% of the recruited patients reported adequate relief of symptoms after consuming a personalized low-FODMAP diet, and the diets did not result in significant reductions in *Bifidobacteria*; symptoms were also significantly alleviated. Other research groups’ findings [98,99] have also confirmed the effectiveness and feasibility of treatment methods involving dividing low-FODMAP diets into three stages, Table 1 has listed major research about how three stages influence microbiota. FODMAP restriction is not the end of the therapy; on the contrary, it is just a starting point. FODMAP restriction should be regarded primarily as a diagnostic approach. If a patient’s symptoms do not improve after a period of treatment, it is advisable to discontinue the restriction immediately and consider alternative treatment methods. Additionally, a low-FODMAP diet is not a healthy diet habit, so it is best for it not to last for too long; only 4 weeks [100] is recommended, and it should not last for up to 12 weeks [101]. After that, if FODMAP restriction works, then the patient can proceed to the next stage—FODMAP reintroduction. The FODMAP reintroduction stage refers to using high-FODMAP food challenges to identify dietary triggers and to test over 3 days at increasing doses. An individual test with one type of food rich in just one FODMAP (such as mango) is employed to ascertain individual tolerance to that entire category of FODMAPs, for instance, fructose [98]. The selected test food should have the capacity to represent all FODMAPs. However, limited research has been published in this regard. Nowadays, examples that are recommended include bread, onions, lentils, milk (lactose), mango (fructose), apricot (sorbitol), and mushrooms (mannitol), along with foods that contain mixtures of FODMAPs [102]. Before each subsequent food challenge, symptoms should ideally be minimal, which can be achieved through a strict FODMAP restriction clearance period of 3 days [103]. The clearance period helps prevent the lingering effects of previous challenges and their impact on symptoms during the next challenge. However, if no symptoms occurred during the previous challenge, the patient can choose the next food type challenge. If a food challenge does not cause any symptoms, then that indicates that the specific food can be regarded as suitable for addition to the patient’s normal diet. The third stage, which is also the most vital component, is FODMAP personalization, which is a continuous self-management period. The third phase involves tailoring the diet to make it more individualized for long-term application, encouraging dietary diversity, meeting nutritional guidelines, and mitigating any negative effects that FODMAP restriction may have on the gut microbiota [104]. The goal of FODMAP personalization is to enhance dietary categories and improve nutritional intake while simultaneously maintaining control over IBS symptoms; it could help patients gradually return to a more normal daily diet. Studies have shown that up to 83% of patients [105] after FODMAP reintroduction and personalization revealed a reduction of more than 50 points on the IBS symptom severity scale, with symptom reduction ranging from 55% to 89%. In addition, dietary nutritional adequacy and FODMAP intake levels returned to baseline habitual levels, leading to improvements in food-related quality of life compared to the baseline [106]. Importantly, compared to patients who returned to a normal diet, those receiving FODMAP personalization did not experience negative impacts on dietary acceptability, food-related quality of life, healthcare utilization, or absenteeism [107]. Therefore, in the long term, FODMAP reintroduction and personalization are important stages for IBS therapy. These two parts could normalize some of the effects of short-term FODMAP restriction, and the adverse effects on *Bifidobacteria* levels caused by FODMAP restriction can be effectively restored through FODMAP personalization, introducing the essential role played by the reintroduction and personalization stages in a low-FODMAP diet. Furthermore, stress and psychological factors have already been identified as triggers for IBS, and diet is related to emotion [108]. Thus, going back to a “normal” diet could alleviate symptoms of IBS. One can employ strategies from both physiological and psychological perspectives. At the same time, the integration of additional adjunctive therapeutic measures might enhance the efficacy of this approach, leading to an improved cure rate and a reduction in patient suffering. For example, yoga has demonstrated effectiveness in improving physical and mental health outcomes for IBS patients [109,110] among different ages, offering a safe and holistic therapeutic approach with promising results. Despite this, hypnosis is another beneficial therapy. According to studies, hypnosis has emerged as a promising treatment for IBS [111,112], showcasing consistent effectiveness in alleviating cardinal symptoms and non-colonic issues. It meets high efficacy standards, offering both physiological and psychological relief mechanisms. Moreover, through establishing biomarker models to accurately predict microbial subtypes [78], we can manage the intake of short-chain fatty acids (SCFAs) in patients with different microbial subtypes and alleviate their pain.

## 7. Conclusions

As is known, IBS is a prevalent condition worldwide, affecting an estimated 10% to 15% of the global population. This chronic gastrointestinal disorder brings about persistent discomfort, including symptoms like abdominal pain, bloating, bowel movement issues, and digestive disturbances, significantly impacting quality of life. Patients may also experience psychological issues such as anxiety and depression, along with social and occupational challenges. Furthermore, IBS poses a burden on healthcare resources, as patients frequently seek medical care for relief. Understanding the prevalence and detrimental effects of IBS is crucial for better management and enhancement of patient quality of life.

In summary, low-FODMAP diets stand out as a significant dietary tool in the management of IBS, providing relief from chronic abdominal pain, bloating, and irregular bowel habits. While the efficacy of these diets in alleviating symptoms is well documented, it is crucial to acknowledge the complex interplay between the low-FODMAP approach and the gut microbiome. Emerging research underscores the potential impact of a low-FODMAP diet on the microbial composition and function within the gut. Evidence suggests that while the diet can bring about symptomatic improvements, it may also lead to unintended consequences such as reductions in beneficial microbial (*Bifidobacteria*) populations. These alterations in the gut microbiota raise concerns regarding the long-term implications for gut health and overall well-being. Although previous studies have primarily focused on the impact of a low-FODMAP diet on the abundance of bifidobacteria in the gut, they have overlooked its effects on other beneficial microbial populations and various microbiome indicators [113]. To address these concerns, strategies such as the reintroduction phase, dietary personalization, and the incorporation of probiotics have been proposed to help restore microbial diversity and promote gut health while maintaining symptom relief. These approaches underscore the importance of individualized dietary management in low-FODMAP diets, recognizing the unique responses of each person’s microbiome to dietary interventions.

Moreover, the findings emphasize the need for further exploration into the intricate relationship between diet, the gut microbiome, and IBS symptomatology. Continued research efforts are essential to elucidate the long-term effects of low-FODMAP diets on the gut microbiome and to refine strategies that optimize both symptom management and microbial balance. By delving deeper into these mechanisms, we can refine our understanding of how dietary choices impact gut health and IBS symptomatology. This nuanced understanding will inform the development of more targeted and sustainable dietary strategies tailored to individual microbiome responses, ultimately enhancing the quality of life for individuals living with IBS. The identification of biomarkers to predict responses to a low-FODMAP diet is of great interest and has become the current research hotspot.

In the future, low-FODMAP diets are supposed to play an important role in the therapy of IBS clinical patients compared to other treatment methods. In addition, their potential applications may extend beyond symptom alleviation to other disease realms such as IBD and gut microbiota dysbiosis. Further research could unveil the comprehensive impact of this dietary approach on health, fostering the development of personalized medicine and offering more effective treatment options for patients.

## Figures and Tables

**Figure 1 nutrients-17-00544-f001:**
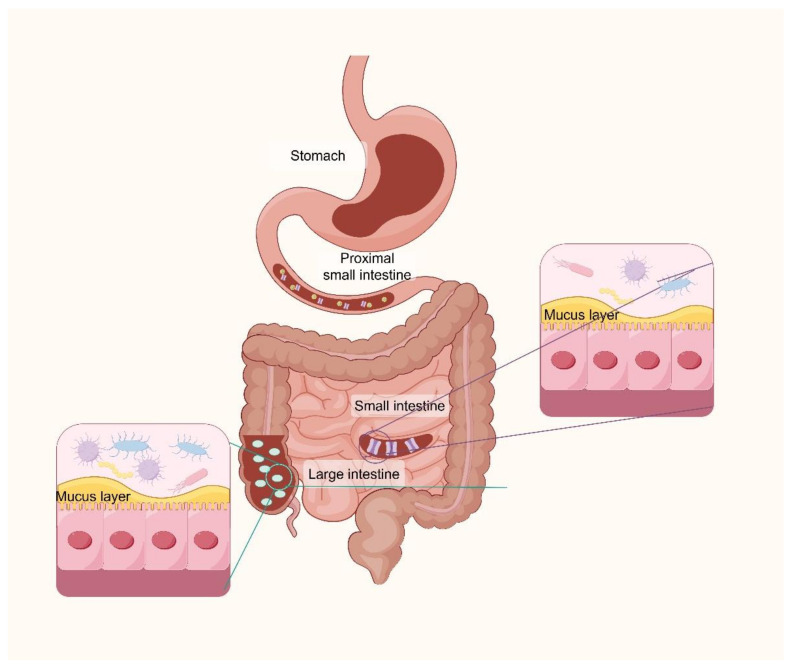
Diagram of the structural composition of the human gastrointestinal tract (created with Figdraw).

**Figure 2 nutrients-17-00544-f002:**
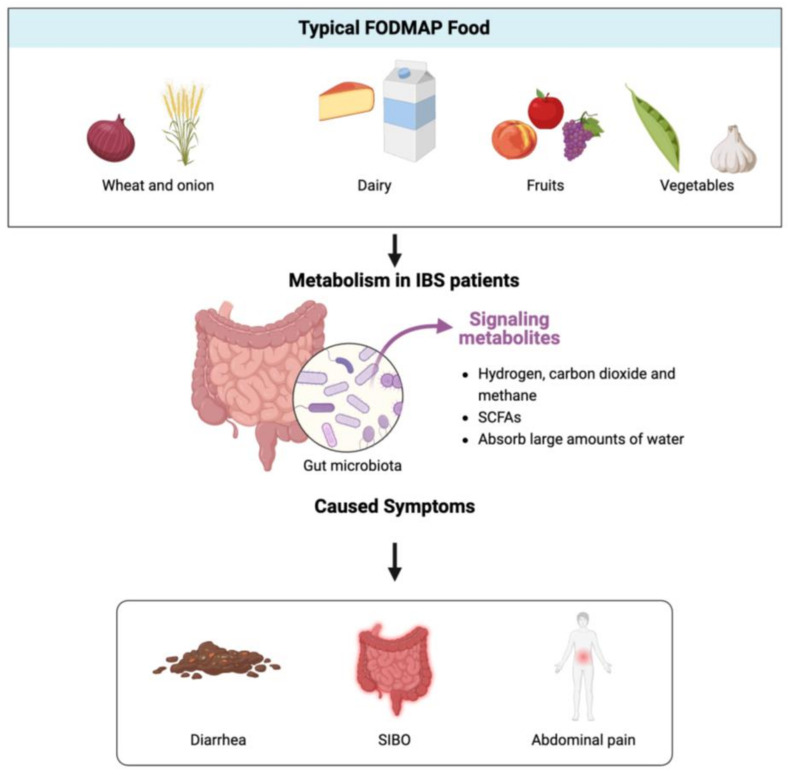
FODMAPs undergo metabolism by intestinal bacteria, leading to the proliferation of harmful bacteria in the gut and causing a series of symptoms in the host (created with biorender).

**Figure 3 nutrients-17-00544-f003:**
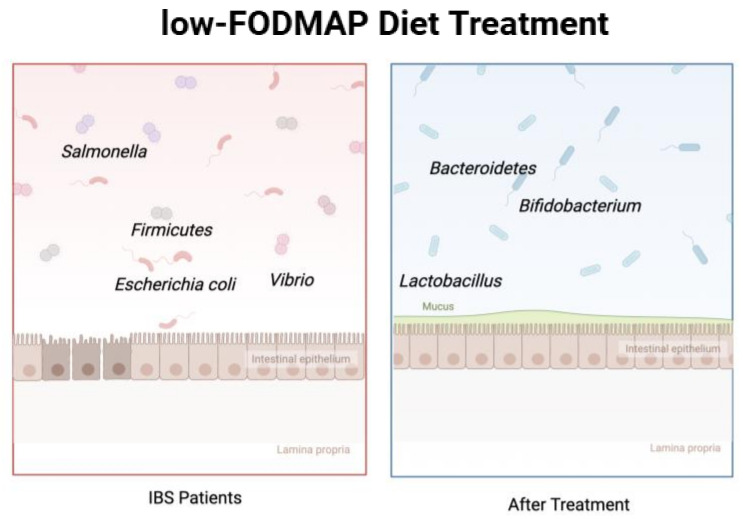
**A** low-FODMAP diet influences the gut microbiota. After treatment with a low-FODMAP diet, there is a significant change in the composition of the gut microbiota in patients, with a decrease in harmful bacteria and an increase in beneficial bacteria (created with biorender).

**Figure 4 nutrients-17-00544-f004:**
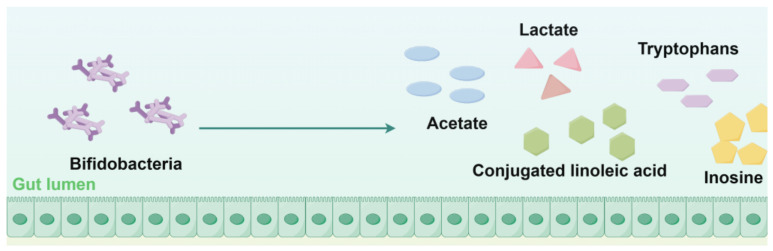
Metabolites produced by *Bifidobacteria* that serve as immune mediators (created with Figdraw).

**Table 1 nutrients-17-00544-t001:** Studies that examined the role of three FODMAP stages and their impact on microbiota.

FODMAP Stages	References	Main Results and Impact on Microbiota
	Staudacher et al., 2017 [86]	Low-FODMAP diet is associated with adequate symptom relief, significantly reduced symptoms, and increased numbers of *Bifidobacterium* species.
Restriction	Whelan et al., 2018 [98]	Patients can adhere to FODMAP personalization, allowing for a less restrictive eating plan that omits their specific FODMAP triggers and promotes a more varied dietary consumption.
	Gibson et al., 2010 [100]	A low-FODMAP diet provides an effective approach to the management of patients with functional gut symptoms. The evidence base is now sufficiently strong to recommend its widespread application.
Reintroduction	Tuck et al., 2017 [102]	The outcome of the re-challenge process aims to find a balance between good symptom control and expansion of the diet.
Personalization	Staudacher et al., 2022 [104]	After 12 months of a personalized low-FODMAP diet, IBS symptoms were significantly alleviated, and levels of *Bifidobacteria* showed no significant difference compared to the baseline.
	Lomer M.C.E., 2024 [105]	Removing the personalized approach and may be less acceptable to patients and may introduce safety concerns in terms of nutritional adequacy.

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
