# Peer review of "Low-FODMAP Diet for Irritable Bowel Syndrome: Insights from Microbiome"

_nutrients, 2025, doi:10.3390/nu17030544_

Round 1
Reviewer 1 Report
Comments and Suggestions for Authors
This is a very interesting review handling an important issue. Not only the symptom relief has to be considered, also other health aspects are of importance since IBS is a harmless disease. Thus, the treatment should not increase other health aspects.
My major conern is that there is no structured Table or figure over the included citattions of the study. The main aim was to review the effect of low FODMAP on microbiota composition. I would like a table that showed all references that have examined microbiota composition during the 3 different phases of low FODMAP, and a short description of the findings in each study. This should improve the manuscript substantially and make it more readable. Now, it is very difficult to get a clear comprehension of the effects
Author Response
Comments 1: This is a very interesting review handling an important issue. Not only the symptom relief has to be considered, also other health aspects are of importance since IBS is a harmless disease. Thus, the treatment should not increase other health aspects.
My major concern is that there is no structured Table or figure over the included citattions of the study. The main aim was to review the effect of low FODMAP on microbiota composition. I would like a table that showed all references that have examined microbiota composition during the 3 different phases of low FODMAP, and a short description of the findings in each study. This should improve the manuscript substantially and make it more readable. Now, it is very difficult to get a clear comprehension of the effects.
Response 1: Thank you for your valuable suggestion. We completely agree that this addition will enhance the clarity and readability of our manuscript. Therefore, we have revised the article to include a comprehensive table that outlines all relevant studies, along with a brief description of their findings.
Reviewer 2 Report
Comments and Suggestions for Authors
This comprehensive review by Zhang and Su focuses on the low-FODMAP diet as a therapeutic approach for individuals with irritable bowel syndrome and also explores how the microbiome is influenced by dietary restrictions of the low-FODMAP diet as well as long-term effects. I have a few specific comments that may help to improve the manuscript:
1. The manuscript text is a bit dense to read. I would suggest summarising data on some key studies in table format, as it may help to improve the readability of the paper and help readers quickly understand the available evidence base.
2. There are a few sentences throughout the manuscript that have language and grammar errors and are difficult to understand. Please check the manuscript for language errors. For example on page 7:
“one former study treated mice with acute vita- 282
min A deficiency leads to a bloom of Bacteroides Vulgatus due to inhibitory effects of 283
retinol on the bacterium, which can be potentially mediated by a decrease in bile acids 284
that inhibit its growth, such as deoxycholic acid, in the deficient-diet-fed mouse.”
3. Abbreviations such as IBS for irritable bowel syndrome are sometimes used and sometimes not throughout the text. I suggest making consistent use of the abbreviations throughout the text.
4. In the conclusion (page 11), the meaning of this sentence is not clear: “In the future, the Low FODMAP diet is supposed to play an important role in the 455
management of IBS.”
Perhaps the authors mean that a low-FODMAP diet may or should play an important role in managing IBS? Please make this sentence clearer.
Comments on the Quality of English Language
2. There are a few sentences throughout the manuscript that have language and grammar errors and are difficult to understand. Please check the manuscript for language errors. For example on page 7:
“one former study treated mice with acute vita- 282
min A deficiency leads to a bloom of Bacteroides Vulgatus due to inhibitory effects of 283
retinol on the bacterium, which can be potentially mediated by a decrease in bile acids 284
that inhibit its growth, such as deoxycholic acid, in the deficient-diet-fed mouse.”
3. Abbreviations such as IBS for irritable bowel syndrome are sometimes used and sometimes not throughout the text. I suggest making consistent use of the abbreviations throughout the text.
Author Response
Comment 1. The manuscript text is a bit dense to read. I would suggest summarising data on some key studies in table format, as it may help to improve the readability of the paper and help readers quickly understand the available evidence base.
Response 1: We have also included a new table summarizing key studies related to the three phases of the low FODMAP diet, highlighting their main findings
2. There are a few sentences throughout the manuscript that have language and grammar errors and are difficult to understand. Please check the manuscript for language errors. For example on page 7: “one former study treated mice with acute vitamin A deficiency leads to a bloom of Bacteroides Vulgatus due to inhibitory effects of retinol on the bacterium, which can be potentially mediated by a decrease in bile acids that inhibit its growth, such as deoxycholic acid, in the deficient-diet-fed mouse.”
Response 2: We sincerely apologize for any language and grammar issues. In response, we have thoroughly reviewed the entire paper and made appropriate revisions to enhance readability and conciseness.
3. Abbreviations such as IBS for irritable bowel syndrome are sometimes used and sometimes not throughout the text. I suggest making consistent use of the abbreviations throughout the text.
Response 3: We have also ensured consistency in the use of abbreviations throughout the text, specifically using "IBS" after its initial definition.
4. In the conclusion (page 11), the meaning of this sentence is not clear: “In the future, the Low FODMAP diet is supposed to play an important role in the management of IBS.” Perhaps the authors mean that a low-FODMAP diet may or should play an important role in managing IBS? Please make this sentence clearer.
Response 4: For this point, we revised the sentence to improve clarity and make it more straightforward.
Reviewer 3 Report
Comments and Suggestions for Authors
Nutritional factors have a significant impact on the profile and metabolism of gut bacteria. However, the results of many studies are not conclusive, which justifies an attempt to analyze and synthesize them. In most studies, a low FODMAP diet has been shown to mainly cause the growth of Bifidobacterium, with small change in the levels of other bacteria. As a result, among other things, the production of SCFAs decreases, which is an unfavorable phenomenon ( Conley T E. Microbiome-driven IBS metabotypes influnce response to the low FODMAP diet: eBioMedicine, 2024,107,105282) , but other researchers do not confirm this relationship ( So D. Effects of a low FODMAP diet on the colonic microbiome in irritable bowel syndrome: a systemic review wit meta-analysis. Am.J.Clin Nutr. 2022,116,943). Numerous studies on the gut microbiome have obtained a variety of results, which makes it difficult to draw conclusions and develop diagnostic and therapeutic recommendations. This overview complements previous similar studies. The problem of treating gut dysbiosis in people with IBS using a persenalized diet requires further metagenomic, metabolomic and clinical research.
The substantive and editorial work of the review does not raise significant comments. The paper may be published, but it is desirable to supplement the references with the above-mentioned publications and to compress the chapter on the pathogenesis and symptoms of irritable bowel syndrome.
Author Response
Comments 1: Nutritional factors have a significant impact on the profile and metabolism of gut bacteria. However, the results of many studies are not conclusive, which justifies an attempt to analyze and synthesize them. In most studies, a low FODMAP diet has been shown to mainly cause the growth of Bifidobacterium, with small change in the levels of other bacteria. As a result, among other things, the production of SCFAs decreases, which is an unfavorable phenomenon ( Conley T E. Microbiome-driven IBS metabotypes influnce response to the low FODMAP diet: eBioMedicine, 2024,107,105282) , but other researchers do not confirm this relationship ( So D. Effects of a low FODMAP diet on the colonic microbiome in irritable bowel syndrome: a systemic review wit meta-analysis. Am.J.Clin Nutr. 2022,116,943). Numerous studies on the gut microbiome have obtained a variety of results, which makes it difficult to draw conclusions and develop diagnostic and therapeutic recommendations. This overview complements previous similar studies. The problem of treating gut dysbiosis in people with IBS using a persenalized diet requires further metagenomic, metabolomic and clinical research.
The substantive and editorial work of the review does not raise significant comments. The paper may be published, but it is desirable to supplement the references with the above-mentioned publications and to compress the chapter on the pathogenesis and symptoms of irritable bowel syndrome.
Response 1: Thank you for your thoughtful feedback on our review. I have carefully reviewed the references you mentioned and have thoughtfully integrated their insights into our manuscript which enriched the discussion and strengthen our arguments. Additionally, I have reevaluated the section on the pathogenesis and symptoms of IBS and made appropriate adjustments to ensure it’s more concise.
Round 2
Reviewer 1 Report
Comments and Suggestions for Authors
Congratulations to a well-written manuscript